# WHAT IS THE ROLE OF MEMORIZATION IN CONTINUAL LEARNING?

## ABSTRACT

Memorization impacts the performance of deep learning algorithms. Prior works have studied memorization primarily in the context of generalization and privacy. This work studies the memorization effect on incremental learning scenarios. Forgetting prevention and memorization seem similar. However, one should discuss their differences. We designed extensive experiments to evaluate the impact of memorization on continual learning. We clarified that learning examples with high memorization scores are forgotten faster than regular samples. Our findings also indicated that memorization is necessary to achieve the highest performance. However, at low memory regimes, forgetting regular samples is more important. We showed that the importance of a high-memorization score sample rises with an increase in the buffer size. We introduced a memorization proxy and employed it in the buffer policy problem to showcase how memorization could be used during incremental training. We demonstrated that including samples with a higher proxy memorization score is beneficial when the buffer size is large.

## 1 INTRODUCTION

Memorization plays an important role in deep learning (Feldman, 2021; Feldman & Zhang, 2020). Deep learning models are capable of memorizing the entire dataset with completely random training labels (Zhang et al., 2017), achieving almost perfect training accuracy. Methods that limit memorization, such as privacy-preserving algorithms (Papernot et al., 2017), fail to achieve classification accuracy comparable to that of standard training methods. Motivated by these observations, later works have shown that achieving high test set performance requires memorization (Feldman, 2021). So far, memorization has been studied primarily through the lens of generalization or privacy (Wei et al., 2024). Continual Learning

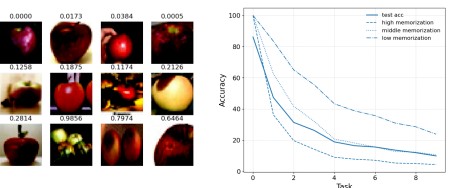

Figure 1: (Left) images with low, middle and high memorization scores from class "apple" from CIFAR100 dataset. (Right) first task accuracy for dataset subsets with different memorization scores. Images with higher memorization scores are forgotten twice as fast as images with low memorization scores.

(CL) (Chen et al., 2018) is the area of machine learning that aims to prevent catastrophic forgetting (French, 1999) in the models trained incrementally with data sampled from different data distributions. There were many methods developed that aim to prevent forgetting (Masana et al., 2022). Counterintuitively, memorization and forgetting prevention are not the same. According to the definition of Feldman (Feldman & Zhang, 2020), memorization refers to the phenomenon where some sample is classified correctly when it is included in the training data. Once removed from the training dataset, it is no longer properly classified. It means that the model can only memorize the sample label without learning any pattern. Such phenomena could occur when the sample has an incorrect label due to a labeling error or when the sample belongs to a long tail (Feldman, 2021) (meaning it could be a minority class in an imbalanced setting or it could contain weakly represented features in the dataset). The opposite of forgetting is not memorization, but complete knowledge retention (Chen et al., 2018). In this case, we are interested in keeping the predictive performance high on past tasks, regardless of

whether the sample belongs to the long tail or not. While forgetting and memorization are different, studying them requires understanding how knowledge is retained in the network. We found that there are relatively few works (Jagielski et al., 2023; Maini et al., 2022; Tirumala et al., 2022b) that study the connection between memorization and forgetting. Moreover, these studies examine forgetting in the context of privacy or label noise, with no focus on the change in data distribution. For this reason, we want to study what is the role of memorization in an incremental training setup, and how Continual Learning is impacted by memorization of training data.

Our key findings could be summarized as follows:

- Increasing the number of classes in dataset increases memorization.
- The higher the memorization score of an example, the greater its susceptibility to forgetting.
- When training with full access to data from past tasks, the classification accuracy of memorized samples remains high.
- A computationally efficient proxy for memorization score can effectively guide buffer policies to improve the performance of incremental training.
- The importance of examples, with a significant proxy memory score, increases for large buffers.

## 2 RELATED WORKS

### 2.1 MEMORIZATION

Memorization (Zhang et al., 2017) in deep learning refers to a phenomenon where deep neural networks learn specific details or particular features of individual training examples, rather than extracting common patterns or generalized features of the underlying data distribution. This can lead to a model's reliance on rote recall, potentially impacting its ability to generalize to new, unseen data and raising concerns about security and privacy (Wei et al., 2024). In (Zhang et al., 2017), it was shown that overparametrized neural networks are capable of fitting training data with random labels with high training accuracy. Further work (Arpit et al., 2017) studied the impact of architecture and dataset size on memorization. It also shows that general concepts are learned early in training, and memorization could lead to more complicated decision boundaries. Anagnostidis et al. (Anagnostidis et al., 2023) showed that even with random labels, deep neural networks learn some features that are beneficial for classification on data with original labels. Further works (Carlini et al., 2019; Maini et al., 2023; Tirumala et al., 2022a) show that memorization does not necessarily lead to overfitting.

In (Feldman, 2021), it was shown that obtaining high accuracy on the test set requires memorization. The atypical samples from the long tail are too few to enable proper representation learning, leading to memorization as the only way of obtaining high performance on the test set. In (Feldman & Zhang, 2020) Feldman et al. introduced an influence score that allows for estimating the influence of training samples on test samples predictions, finding that there are training examples that impact the correct classification of unseen long-tail data.

So far, there is no consensus among researchers on what parts of the neural network are responsible for memorization (Wei et al., 2024). Some evidence (Anagnostidis et al., 2023) suggested that the last layers are used for memorization. Other studies (Maini et al., 2023) indicated that neurons responsible for memorization are scattered across the whole network. In (Feldman & Zhang, 2020), it was shown that a classifier is not responsible for memorization, suggesting that memorization is a phenomenon that concerns primarily representation learning.

### 2.2 CONTINUAL LEARNING

Continual learning addresses the challenge of training models on a sequence of tasks with differing data distributions, rather than on a single i.i.d. dataset (Chen et al., 2018). A major issue in this setting is catastrophic forgetting, where performance on earlier tasks sharply deteriorates as the model learns new ones (French, 1999). To tackle this, continual learning methods are generally grouped into three categories. Regularization-based methods reduce forgetting by constraining changes to important parameters. Elastic Weight Consolidation (EWC) adds a regularization term to penalize

updates to critical weights (Kirkpatrick et al., 2016). Learning without Forgetting (LwF) maintains previous knowledge by using pseudo-labels from earlier task classifiers (Li & Hoiem, 2018). FeTrIL (Petit et al., 2023) applies a pseudo-feature generation strategy with the usage of a frozen backbone. Magistri et al. (Magistri et al., 2024) address feature drift by regularizing direction relevant for past tasks.

Rehearsal-based methods rely on memory buffers to replay examples from previous tasks (Chaudhry et al., 2019b). Gradient Episodic Memory (GEM) and its efficient variant aGEM, constrain gradient updates to prevent loss on earlier tasks (Chaudhry et al., 2019a; Lopez-Paz & Ranzato, 2017). More recent work addresses the limitations of small memory buffers using asymmetric updates and classifier corrections (Chrysakis & Moens, 2023). Other strategies, like Dark Experience Replay (DER), store model logits alongside data and use them in a distillation loss to preserve knowledge (Boschini et al., 2023; Buzzega et al., 2020a). Buffer policy algorithms select what samples should be stored in the buffer (Hao et al., 2023; Tiwari et al., 2022; Tong et al., 2025). Researchers used bilevel optimization (Hao et al., 2023; Tong et al., 2025), gradient approximation (Aljundi et al., 2019; Tiwari et al., 2022), or classification uncertainty (Bang et al., 2021) to select samples stored in the buffer.

Expansion-based methods adapt the model architecture to accommodate new tasks. Progressive Neural Networks (PNNs) add task-specific subnetworks that reuse prior knowledge through lateral connections (Rusu et al., 2016). Other approaches expand network parameters with selective retraining to retain performance on old tasks (Yoon et al., 2018). Some methods further enhance this by introducing additional convolutional features and training with a specialized loss function to encourage diverse representations for new data (Yan et al., 2021).

### 2.3 MEMORIZATION IN CONTINUAL LEARNING

Memorization in Continual Learning was studied primarily through the lens of privacy (Ozdenizci et al., 2025; Tobaben et al., 2025). In (Desai et al., 2021), Differential Privacy for Continual Learning was proposed. This algorithm uses a data sampling strategy and moment accountant to provide formal privacy guarantees across tasks, achieving tighter privacy loss while maintaining model utility. Authors of (Tobaben et al., 2025) use a prototype classifier with adapters to ensure that the pretrained model can be continually trained with improved privacy. Prior works have evaluated the connection between forgetting and memorization (Jagielski et al., 2023; Maini et al., 2022; Tirumala et al., 2022b). However, this research was conducted on data with stationary distribution.

## 3 METHODS

### 3.1 NOTATION AND SETTING

In continual learning, a neural network is trained on a sequence of tasks with different data distributions. Each task $t$ is defined by a dataset $D_t = \{(x_i, y_i)\}_{i=0}^{n_t}$, where $x_i$ is image, $y_i$ is label, and $n_t = |D_t|$. We focus on the Class-Incremental Learning setting (van de Ven & Tolias, 2019), where each task consists of a disjoint set of classes. The model $f$ with parameters $\theta$ is trained sequentially by minimizing the loss on the current task: $\mathcal{L}(f(\theta), D_t)$, with access only to the most recent data. Rehearsal-based methods maintain a buffer $\mathcal{M} = \{(x_j, y_j)\}_{j=0}^{m}$, where $m \ll n_t$, to store selected examples from previous tasks. To avoid ambiguity, we refer to training on the full dataset as *stationary training* (with a fixed data distribution), and to sequential task training as *incremental training*.

#### 3.1.1 MEMORIZATION SCORE

There are many possible definitions of memorization proposed in the literature (Wei et al., 2024). This work adopts the Feldman (Feldman, 2021) definition based on memorization score:

$$mem(i, A) = E_{f \sim A(D)}[P(f(x_i) = y)] - E_{f \sim A(D/x_i)}[P(f(x_i) = y)] \tag{1}$$

where $A$ is a training procedure containing randomness that produces a trained network $f$. Memorization score is the difference between the probability assigned to the correct label by the model trained on the whole dataset and the probability assigned by a model with the $i$-th sample removed from the dataset. Such a definition allows for the detection of memorized samples, but it requires

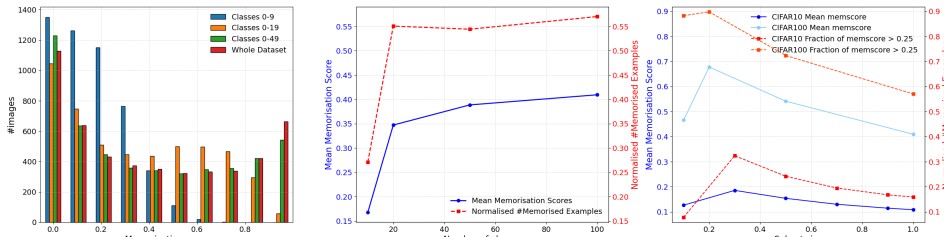

Figure 2: The impact of data and architecture on memorization scores. (Left) histogram of memorization scores for different number classes in the training dataset. (Middle) the mean memorization score and number of samples with memorization score above 0.25 for different number of classes in training dataset (Right) the mean memorization score and number of samples with memorization score above 0.25 for different dataset subset size.

training multiple networks for each learning example to compensate for randomness in $A$. For this reason, Feldman et al. (Feldman & Zhang, 2020) introduced an estimator that reduces the required computational burden, defined as:

$$mem_k(i, A) = E_{f \sim A(\mathcal{S}_i)}[P(f(x_i) = y)] - E_{f \sim A(\mathcal{S}_{-i})}[P(f(x_i) = y)] \qquad (2)$$

where $\mathcal{S}_i$, $\mathcal{S}_{-i}$ are two random subsets of $D$, each of size $k$, with $S_i$ containing $x_i$ and $S_{-i}$ excluding $x_i$. Feldman showed that this estimator's variance is bounded by a function of $k/n$ (where $n = |D|$) and the number of networks trained, $u$. To keep variance small in our experiments we choose $k/n = 0.5$ and $u = 250$. According to Feldman bound (Feldman & Zhang, 2020), such a setup ensures that $mem_m$ variance is below 0.016. We study the accuracy of this estimator in more depth in Appendix C. Unless stated otherwise, we use reduced ResNet18 model (He et al., 2016), tailored to CIFAR100 (Krizhevsky, 2009; Lopez-Paz & Ranzato, 2017) dataset. Overall, for the next part of the experiments, we trained over 3500 neural networks. The exact hyperparameters used for training are given in the Appendix D. When we average the results, we repeat runs with different random seeds, and report standard deviation in tables or as error bars in plots. From now on when referring to memorization score we will mean Feldman estimator.

### 3.2 IMPACT OF TASKS SPLIT ON MEMORIZATION

We begin by analyzing how splitting the data into several tasks and changing the number of classes affects the proportion of training samples with high memorization scores. For this purpose, we train ResNet18 (He et al., 2016) on CIFAR100 (Krizhevsky, 2009) subsets with the first 10, 20, 50, and 100 classes and compute memorization scores for the samples from the first 10 classes. The histogram of memorization scores is presented on the left side of Fig. 2. As the number of classes gets smaller, the memorization scores are lower. This is most evident for score close to one, when full dataset gets largest number of samples. In the middle pane of the Fig. 2 we plot mean memorization scores and number of samples with memorization scores above 0.25 for different number of classes in training set. It shows clear decline in the mean memorization score.

Previous results from the literature suggest that reducing dataset size should increase memorization (Arpit et al., 2017; Li et al., 2024). By reducing the number of classes, we also limit the number of available training examples. To decouple these two effects, we check the impact of dataset size on memorization. To this aim, we evaluate memorization scores for different subsets of CIFAR100 with 0.1, 0.2, and 0.5 of the original dataset size. We keep the number of classes equal to 100, and the images are sampled with stratification to keep the class balance. Lowering the number of learning examples in dataset with 100 classes can lead to low number of samples per class. For this reason we also carry our analogous experiments with different subsets of CIFAR10 (Krizhevsky, 2009) with 0.1, 0.3, 0.5, 0.7, and 0.9 of the original dataset size. In the right panel of Fig. 2 we plot mean memorization scores and fraction of the dataset with memorization scores above 0.25. Our results are consistent with prior works, suggesting that the number of classes in the dataset has a stronger influence than a reduced number of training samples. One can explain such a phenomenon by the

limited capacity of the network. Models trained with a smaller set of classes can learn features that are tailored well to training data, alleviating the need for excessive memorization. If we introduce additional classes into the dataset, while keeping the capacity fixed, the model will have to learn more general features that generalize well across many classes. This interpretation is in line with results from (Harun et al., 2024b), where it was shown that a higher number of classes positively correlates with Out-Of-Distribution performance. This means that the model will have to memorize more samples, as the general patterns will not cover more specific samples from all classes. To verify our conclusions, we provide in Appendix E the memorization scores evaluated for 20 and 50 classes, showing very similar trends as for the first 10 classes. We also check if the same trend could be observed in other datasets. We repeat our experiments for the TinyImageNet dataset (Wu, 2017) and find a very similar trend (please refer to Appendix F for results).

**Impact of architecture.** The impact of architecture on memorization and Continual Learning has already been studied in the literature. In (Arpit et al., 2017), it was shown that wider networks fit noisy data better. In the context of Continual Learning, studies have found that wider models tend to forget less, while deeper models exhibit greater forgetting (Guha & Lakshman, 2024; Mirzadeh et al., 2022). Motivated by these findings, we also investigate how model depth and width affect memorization scores. First, we compute memorization scores for ResNet34 and ResNet50 trained on the entire Cifar100 dataset and plot memorization histograms in Fig. 3 (top). As the network depth increases, the memorization scores decrease significantly. In Fig. 3 (bottom), we plot memorization scores for ResNet18 trained on full Cifar100 with different width multipliers. With the increase in model width, the mean memorization score quickly saturates, but the increase is visible in the samples with the highest memorization score. Therefore, some parallel exists between wider models that are less prone to forgetting and exhibiting more memorization, but the connection is not strong.

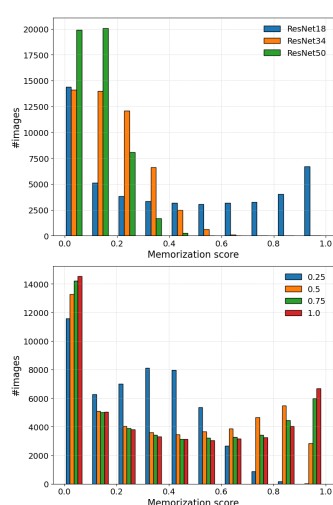

Figure 3: Memorization scores histograms for different model architectures (top) and model widths (bottom).

### 3.3 INCREMENTAL TRAINING

The Feldman formulation of memorization score (Feldman & Zhang, 2020) is difficult to apply in the Continual Learning scenario. The exclusion of training examples from one task would impact representation learning in the following tasks, leading to a combinatorial explosion of possible training data arrangements that need to be considered when evaluating memorization. To circumvent this problem, we instead use memorization scores computed offline for the whole dataset and track how well samples with higher memorization scores are classified throughout the whole incremental training. We employ a threshold of 0.25 used in (Feldman & Zhang, 2020) to select memorized samples. We study three scenarios. First, we use standard experience replay with reservoir sampling and a buffer of size 500 for Split-Cifar100. As shown on the left-hand side of Fig. 4, the accuracy for memorized samples drops significantly after training on two subsequent tasks. Afterward, the accuracy declines at a similar rate for both the test set samples and the training examples with high memorization scores. In the second setting, we study training with unlimited access to past tasks' memory (Fig. 4 middle left). During training with an infinite buffer, the accuracy for memorized samples slowly decreases but remains high over the course of the entire training process. This is in line with Feldman's observation about memorization being beneficial and necessary for high performance (Feldman, 2021). It is also in line with our previous results for memorization with a different number of classes in the training set. As we introduce new classes, the model needs to learn more general features, gradually increasing the number of samples that need to be directly memorized.

Lastly, we study the incremental training with a method that does not use a buffer. We choose Learning without Forgetting (Li & Hoiem, 2018) (LwF) due to its popularity. On the middle right-

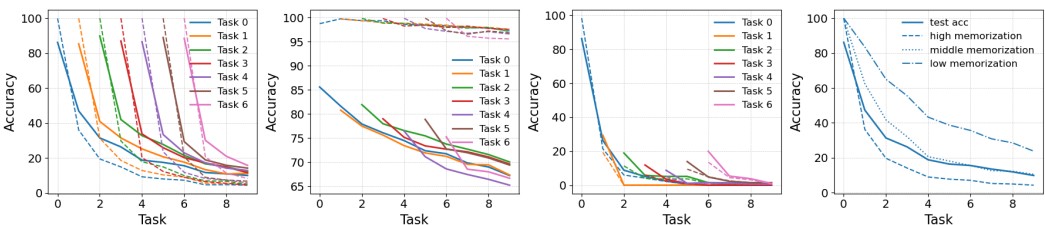

Figure 4: Task accuracy for test set (solid line) and samples with high memorization (dotted line) across incremental training on Seq-Cifar100 stream with 10 tasks. (Left) training with buffer size 500. (Middle left) training with full access to previous tasks. (Middle right) training with LwF. (Right) Accuracy on task 0 when training with buffer size 500 for data with different memorization scores. Results averaged over 5 runs.

hand side of Fig. 4, we see that the accuracy for memorized samples also decreases significantly after the data distribution change. For newly introduced tasks, the accuracy of memorized learning examples is lower than the test set accuracy. In previous scenarios, the initial accuracy for memorized samples was close to 100%. It shows that learning with heavy regularization affects plasticity, reducing the model's capability to classify data with high memorization scores well. On the other hand, the accuracy for memorized samples does not drop entirely to zero instantly, which suggests that some trace of memorized data remains in the network, even without access to past data. In Appendix I, we include results with different thresholds for determining memorized samples and show that a change in threshold value does not affect our conclusions significantly. To complete the analysis, we also provide the accuracies for dataset subsets with different memorization scores in Fig. 4. As in the previous part of our experiments, we define samples with high memorization as the ones with a memorization score above 0.25, middle memorization between 0.1 and 0.25, and low memorization below 0.1. With CIFAR100, it means that 28058 out of 50000 training set samples have been identified as those with a high memorization score (please refer to the left side of Figure 1 for verification). It means that half of the training set is forgotten at a faster pace than the other.

### 3.4 MEMORIZATION SCORE PROXY

Determining the memorization score is compute-intensive, even using Feldman estimator (Feldman & Zhang, 2020). For this reason, it is not feasible to use the memorization score directly during incremental training. To circumvent this issue, we introduce a proxy that approximates the memorization score and is more accessible during training. We base our proxy on the observation made in (Arpit et al., 2017), namely that learning of patterns takes place in early stages of training, while memorization is prone to happen in the later stages of learning. The same premise was used to detect noisy labels in (Maini et al., 2022). During offline continual training, we can store the first iteration when the given sample was classified correctly, and later, the prediction did not change in the following epochs. To be more specific, we define our proxy as:

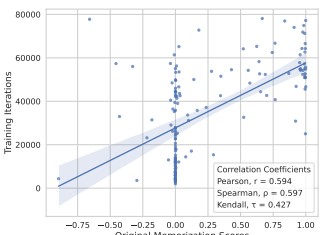

Figure 5: Correlation of training iteration with memorization score.

$$mem_{proxy}(i, A) = min\{j \in \mathbb{N} | f_j(x_i) = y_i \land \forall_{k>j} f_k(x_i) = y_i\} \qquad (3)$$

where $f_j$ is the model before the gradient update iteration $j$. In practice, evaluating each sample in every iteration would add a huge overload. For this reason, we check if current predictions match the ground truth after the forward pass with the current minibatch. This approach is straightforward and adds minimal computational overhead. As shown in Fig. 5, the correlation between training iteration and original memorization score is moderate, however, we show in Appendix C that it correlates well with the Feldman estimator.

## 3.5 Memorization-aware Experience Replay

We want to examine how memorization could be used during incremental training. In particular, we are interested in the buffer policy problem - deciding what samples should be stored in the buffer. Storing samples with different memorization scores could impact the overall performance differently. We compute a proposed memorization score proxy to select samples that should correspond to low, medium, or high memorization and use them to update the buffer. The full algorithm for our approach is given in Appendix J.

We are using reservoir sampling and replacing buffer content with selected top-k, mid-k, or bottom-k samples at the end of training. We found in preliminary experiments that such an approach works better than updating the buffer only at the end of task training. During the buffer update, we replace the samples from the current task that were already stored in the buffer. We use a balanced variant of reservoir sampling to ensure that the data in the buffer is always balanced.

## 4 Experimental setup

**Datasets.** Our experiments follow the class-incremental learning scenario (van de Ven & Tolias, 2019), using standard continual learning benchmarks created by splitting datasets into multiple tasks. Specifically, we use CIFAR-10, CIFAR-100 (Krizhevsky, 2009), and Tiny ImageNet (Wu, 2017), divided into 5, 10, and 20 tasks, respectively. The order of classes across tasks is randomized using different seeds.

**Metrics.** We use two evaluation metrics, namely the final test set accuracy averaged over all tasks, defined as $Acc = \frac{1}{K} \sum_t^K \frac{1}{n_t} \sum_{i=1}^{n_t} \mathbb{1}[f(x_i, \theta_K) = y_i]$, where $K$ is the number of tasks, and $\mathbb{1}$ is an indicator function and forgetting measure (FM) (Chaudhry et al., 2018) defined as average difference between maximum obtained accuracy, and final accuracy for given task.

**Baselines.** We use standard reservoir sampling (Chaudhry et al., 2019b), balanced reservoir sampling from (Buzzega et al., 2020b), Rainbow Memory (Bang et al., 2021), Bilevel Coreset Selection (BCSR) (Hao et al., 2023), and Probabilistic Bilevel Coreset Selection (PBCS) (Zhou et al., 2022) as baselines in our experiments. For BCSR and PBCS, we use a small auxiliary convolutional model for sample selection, which consists of two convolution layers, a max pool between convolutions, and two linear layers.

**Implementation.** For all rehearsal-based methods, we use a buffer of size 500 unless specified otherwise. When available, we adopt the best hyperparameters reported by the original authors; otherwise, we use the settings detailed in Appendix D. All experiments are implemented using the Mammoth library (Buzzega et al., 2020a). We made our code available online [1].

## 5 Results

### 5.1 Evaluation with standard benchmarks

First, we carry out an experiment on a standard set of benchmarks used in Continual Learning. We compare accuracy and FM for the proposed approach with other buffer policy methods in Tab. 1. In the standard setup, we can see that selecting lower and mid proxy memorization scores obtains better results than reservoir sampling. This suggests that with the lowest memory budgets, retaining the performance for standard data is challenging enough, therefore we should be constructing a buffer with the most typical samples that are easy to learn and represent well, given the class. This shows that in such a setting, memorization does not play a significant role as forgetting prevention is more important, however the memorization could be used to guide the buffer construction process.

We acknowledge, that our results may lack some of the baselines, that may obtain better performance that the proposed method like (Aljundi et al., 2019; Tiwari et al., 2022; Tong et al., 2025), however our primary motivations is not to propose the best algorithm for buffer policy, but rather study the impact of memorization on Continual Learning. We also show in Tab. 2 that the proposed approach also works with other rehearsal algorithms (Buzzega et al., 2020a; Caccia et al., 2022).

---

[1]`https://anonymous.4open.science/r/memorization-cl-8CDA/README.md`

Table 1: Accuracy for incremental training on Split-Cifar10, Split-Cifar100 and Split-TinyImageNet benchmarks. The results averaged over 5 runs.

| buffer policy | Split-Cifar10 | | Split-Cifar100 | | Split-TinyImageNet | |
|---|---|---|---|---|---|---|
| | Acc(↑) | FM(↓) | Acc(↑) | FM(↓) | Acc(↑) | FM(↓) |
| reservoir | 55.18±6.59 | 42.45±7.04 | 22.07±0.84 | 65.47±0.62 | 6.38±0.62 | 75.87±0.54 |
| reservoir balanced | 55.25±4.66 | 42.35±4.94 | 22.39±0.75 | 65.01±0.20 | 6.53±0.58 | 75.99±0.68 |
| rainbow memory | 56.86±3.17 | 39.91±4.07 | 23.19±0.79 | 64.16±0.48 | 6.42±0.52 | 76.07±0.73 |
| BCSR | 55.83±4.71 | 41.37±5.69 | 22.84±0.84 | 64.80±0.53 | 6.65±0.52 | 75.86±0.43 |
| PBCS | 55.22±5.31 | 42.35±5.44 | 23.14±1.09 | 64.39±0.89 | 6.93±0.57 | 75.79±0.42 |
| bottom-k memscores | 56.04±4.80 | 41.80±5.25 | 25.88±1.14 | 61.60±0.71 | 7.85±0.29 | 74.87±0.54 |
| middle-k memsocres | 55.96±4.62 | 41.80±4.98 | 23.62±1.07 | 63.81±0.54 | 7.11±0.57 | 75.52±0.85 |
| top-k memscores | 41.16±7.38 | 56.60±7.85 | 17.28±1.19 | 69.95±0.87 | 5.18±0.13 | 77.12±0.63 |

Table 2: The Accuracy and Forgetting Measure for the ER-ACE and DER++. The results averaged over 5 runs.

| method | Split-Cifar100 | | Split-TinyImageNet | |
|---|---|---|---|---|
| | Acc(↑) | FM(↓) | Acc(↑) | FM(↓) |
| ER-ACE | 36.81±0.86 | 34.02±0.36 | 15.07±0.35 | 39.68±1.17 |
| +bottom-k | 40.03±0.29 | 32.18±1.2 | 19.27±0.81 | 35.59±0.76 |
| +mid-k | 38.69±0.76 | 31.29±1.08 | 15.76±0.52 | 39.43±1.55 |
| +top-k | 33.22±0.69 | 37.84±1.54 | 12.92±0.37 | 42.07±1.42 |
| DER++ | 33.9±1.67 | 50.12±2.16 | 12.5±0.94 | 58.29±1.99 |
| +bottom-k | 38.34±1.40 | 45.13±1.42 | 15.09±0.81 | 55.14±1.88 |
| +mid-k | 34.75±2.37 | 49.04±2.51 | 13.04±1.26 | 57.80±2.27 |
| +top-k | 28.32±2.92 | 56.9±3.07 | 7.60±1.10 | 66.79±2.57 |

Table 3: Test accuracy for buffer policies based on mixing high memorization samples with mid and bottom on Seq-Cifar100 and various buffer sizes.

| method | buffer size | |
|---|---|---|
| | 2000 | 5000 |
| bottom-k memscores | 42.24±0.45 | 52.64±0.61 |
| +10% top-k | 42.17±0.3(-0.07) | 52.46±0.62(-0.18) |
| middle-k memsocres | 40.52±0.57 | 52.75±0.71 |
| +10% top-k | 40.77±0.64(+0.25) | 52.74±0.43(-0.01) |
| method | 10000 | 20000 |
| bottom-k memscores | 59.56±0.69 | 64.81±0.87 |
| +10% top-k | 59.61±0.43(+0.05) | 64.91±0.68(+0.10) |
| middle-k memsocres | 59.88±0.52 | 64.57±0.45 |
| +10% top-k | 60.16±0.48(+0.28) | 64.93±0.64(+0.36) |

## 5.2 Training with larger buffers

We evaluate how the classification performance of proposed policies changes with an increase in buffer size. In Fig. 6, we plot test set accuracies for different buffer policies based on proxy memorization score. Regardless of buffer size, selecting samples with the lowest or medium memorization performs better than random selection. Also, as the buffer size increases, the performance obtained for selecting top-k samples improves. This suggests that keeping hard samples in the buffer is necessary to obtain performance close to the upper bound.

To further evaluate the impact of high memorization in different regimes, we modify the sample selection process for bottom-k and mid-k to reserve 10% of the current task buffer space for samples with the highest proxy memorization score. The results are presented in Tab. 3. As in the previous case, the benefit of including samples with high proxy values is mostly visible for higher buffer sizes. However, the increase in accuracy is lower than the standard deviation, suggesting that the impact is low.

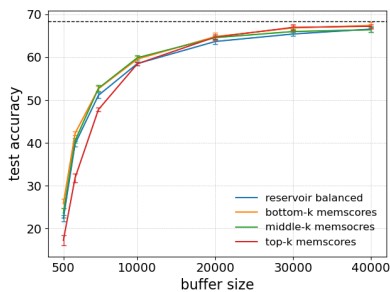

Figure 6: Test set accuracy for various buffer sizes on Split-Cifar100 benchmark. Results are averaged over 5 runs. The black horizontal line denotes training with full access to memory. The results averaged over 5 runs.

## 6 Discussion

Our results suggest that as the number of classes in the dataset increases, the representations learned by the network become more general. At the same time, larger fractions of data are not being covered by general patterns should be memorized. We can think of results for each number of classes as the upper bound of performance at different stages of incremental training. It shows the changes in the

memorization as we progressively increase the number of tasks. Increasing memorization could be especially challenging for exemplar-free CL methods, which do not have access to prior labels and keep model capacity fixed. This means that, in principle, Continual Learning with either increasing architecture (Rusu et al., 2016) or rehearsal with memory (Buzzega et al., 2020a; Chaudhry et al., 2019b) should be much easier.

In our experiments we found that deeper models promote less memorization, while wider models exhibit larger memorization. In the Continual Learning, the trend is reversed, namely: deeper models forget more, while wider models forget less. Similarly, we found that larger values of weight decay lower the memorization, while the Continual Learning algorithms based on rehearsal use low or no weight decay regularization Buzzega et al. (2020a); Boschini et al. (2023); Harun et al. (2024a); Mirzadeh et al. (2020). These results suggest that larger memorization is actually beneficial for the incremental learning process. On the other hand, we found that memorized samples are forgotten at significantly faster rates than standard learning examples. In such case increasing memorization should be detrimental for the stability of the model, and improve only plasticity by allowing to obtain higher test set performance on current task Feldman (2021). These findings are contradictory to each other. Unraveling this conundrum requires further investigation, both theoretical and empirical, into the role of memorization in the Continual Learning process.

The accuracy obtained for selecting samples with the top proxy memorization score increases as the buffer grows. This indicates that when the memory size limitation is relaxed (as it is postulated by several recent Continual Learning works (Harun et al., 2024a; Knoblauch et al., 2020; Peng et al., 2022)), the importance of memorized samples is increasing. We believe that progress in incremental learning can be achieved by studying both training with high access to memory and low or no memory access. The first facilitates the necessary conditions for achieving high performance, that is, comparable to full access to memory, while the second provides highly efficient methods in terms of memory usage. Practitioners later could select methods from both ends of this spectrum to solve specific problems they work on.

**Limitations** Memorization could be affected by the training procedure and hyperparameter selection. Factors that are known to impact memorization include data augmentation (Li et al., 2024), regularization (Zhang et al., 2017; Arpit et al., 2017), and data repetition (Zhang et al., 2023). In this study, we focused on popular Continual Learning setups (Buzzega et al., 2020a), but additional studies are needed to assess the impact of other factors. Moreover, Feldman et al. (Feldman & Zhang, 2020) showed that CIFAR100 has some samples that occur both in training and in the test set, leading to increased influence of some atypical samples on the test set. We tried to address that by using different datasets and settings (see Appendices E,I, and F). However, at the same time, CIFAR100 is one of the most popular datasets used for studying Continual Learning. Future experiments should include broader spectra of datasets to obtain more robust results. Finally, we do not solve the problem of determining exactly what samples are memorized during incremental training. We leave it for future research, but we acknowledge that developing such a method could shed more light on the role of memorization in Continual Learning.

## 7 CONCLUSION

Memorization is a necessary component for achieving a classification performance comparable to training with full memory in Continual Learning. Yet, simultaneously, the capability to correctly classify the samples with high memorization scores drops significantly after a change in data distribution. Our experiments with standard CL benchmarks show that at lower memory regimes, forgetting of regular data is a more important consideration than forgetting of long-tail data. Regardless, we show that the notion of memorization can still be useful in constructing buffer policy, even if data with a high memorization score is not important. Our further experiments show that closing the gap between stationary and incremental training requires taking memorization into consideration.

Future work could include the localization of which parts of the network are responsible for memorization (Anagnostidis et al., 2023; Maini et al., 2023) and designing a proper measure to protect these parts from forgetting in incremental training. We know what factors affect memorization in stationary training. However, the factors that could impact memorization in incremental learning are unknown. We believe that studying these factors could also be beneficial.

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

## A  ETHICS STATEMENT

Our work primarily studies the properties of continual learning algorithms in the context of their performance; therefore, it is hard to predict how presented results will impact particular applications, however we find that two key aspects require attention. Firstly, our work shows that memorization needs to be considered when designing well-performing incremental training algorithms. This claim has a significant impact on data privacy, as it puts responsibility on the user of the incremental-training algorithm, which creates a large buffer to store past data. This user must curate the data used for rehearsal. This process should involve ensuring that the data owner has agreed to particular data usage and that the data is properly licensed and anonymized. Failure to adjust to these requirements can lead to leakage of private or proprietary information used for training.

The second consideration is the higher computational cost compared to other experiments. Evaluating the memorization requires more computational power due to the repeated training process, which comes at increased environmental costs due to carbon emissions. However, the incremental learning has the potential to reduce the required training time by removing the need for training from scratch every time new data with a different distribution arrives. We believe that our experiments can potentially reduce the cost of updating deep learning models in the future, at the expense of larger power consumption now.

## B  THE USE OF LARGE LANGUAGE MODELS

During preparation of this paper Large Language Models (LLM) were used in two areas, namely: text edition and code writing. In the case of text edition, all of the text in this paper was written by humans, the LLMs were used only to improve spelling, grammar and overall text edition. In the case of code writing the LLMs were used to aid writing the code for visualization for this paper. All the code for the experiments was written by humans.

## C  MEMORIZATION SCORE PROXIES

The original memorization scores defined in (Feldman, 2021) require several neural network trainings to compute a score for only a single sample. For this reason, we analyzed several possible proxies that require less computation and could be applied more easily in our experiments. Additionally we wanted to check if proposed method produces substantially different scores for samples, compared to other methods from the literature. We have considered the following proxy scores for memorization:

- **Feldman estimator** - introduced in (Feldman & Zhang, 2020), the estimator limits the number of trainings required to compute the memorization scores. If the original memorization score can be defined as leave-one-out, the estimator is leave-k-out. It requires less computation compared to the original memorization score, however, it is still not usable during incremental training.

- **Cosine distance** - For each sample, we compute the cosine distance between its feature representation and the mean feature vector of the class to which the sample belongs in the learned latent representation space. It measures the angular distance between vectors, providing a normalized metric that is invariant to the magnitude of the feature representations. A higher distance implies the sample is more dissimilar to the prototypical class member and potentially more difficult or atypical. This approach follows the distance-based rehearsal policy used in GRASP (Harun et al., 2024a), which progressively selects more diverse or hard-to-learn samples from the class center using cosine distance.

- **Mahalanobis distance** - we compute the Mahalanobis distance in the learned latent representation space of the neural network. We use a feature vector representing the given sample and the mean and covariance of the class to which the sample belongs. A similar solution for rehearsal policy was introduced in (Harun et al., 2024a).

- **Euclidean distance (L2 norm)** - this metric represents the straight-line distance between the feature vector representing the given sample and the mean feature vector of the class to which the sample belongs in the learned latent representation space. It has been widely used in continual learning for class-mean-based rehearsal strategies to select samples closest

to class centers (Castro et al., 2018; Chaudhry et al., 2019b), where samples closest to the mean are considered representative.

- **iCaRL ranks** - we compute the ranking of samples based on the herding algorithm introduced in iCaRL (Rebuffi et al., 2017). The herding method selects an exemplar set in a fixed order that minimizes the distance between the mean of selected exemplars and the class mean in the representation space. In the first order, the algorithm selects the samples that best approximate the mean of the given class. However, this algorithm does not directly provide a numerical score for each sample, which is necessary for evaluating correlation with our memorization score. For this reason, we converted the order in which samples are selected into the ranks, meaning that the first selected sample is assigned rank 0, the second sample rank 1, and so on. Lower ranks indicate samples that would be prioritized for selection in the herding-based rehearsal strategy. These ranks are then used as a proxy score, where lower ranks correspond to samples that are closer to the class feature mean.

- **LASS distance** - in (Arpit et al., 2017) the Langevin adversarial sample search (LASS) was introduced to find adversarial samples that are in some predefined $||L||_\infty$ neighborhood of the sample. The authors used this method to study how complicated the decision boundary is and found that neural networks trained with random labels have much more complicated decision boundaries compared to standard training. Here we reuse the LASS algorithm, however, we do not limit the neighborhood size. Instead, we set it to some large constant and use distance from the original sample $||x_{adv} - x||_\infty$ as the proxy.

- **Carlini-Wagner distance** - the LASS algorithm was designed primarily to better explore the search space and not to find the minimal perturbance that causes a change in classification. For this reason, we use an untargeted Carlini-Wagner adversarial attack (Carlini & Wagner, 2017), that directly optimizes for the smaller perturbance in the $||L||_2$. We utilize this algorithm to generate the adversarial example, and then compute the $||x_{adv} - x||_2$ difference and use it as a proxy.

- **Training iteration** - based on previous results (Arpit et al., 2017; Maini et al., 2022) we can assume that simple patterns are trained in the first epochs, while memorization mostly happens later. For this reason, we may consider the iteration at which the sample was classified correctly (and was classified correctly until the end of the training) as a valid proxy for memorization score.

To evaluate how closely the proxies match the original memorization scores, we compute the original memorization score for randomly sampled 150 samples from CIFAR100, and then measure the correlations between each proxy and the memorization score (see Fig. 7). To limit the computation used during the calculation of the original memorization score, we train only a single network for a single sample excluded from the dataset and a single network for the whole dataset.

For each correlation, we report Pearson $r$, Spearman $\rho$, and Kendall $\tau$. Pearson's $r$ quantifies the linear relationship between two variables. Spearman's $\rho$ is a rank-based measure of monotonic association, reflecting how often the ordering by proxy matches the ordering by the original score. Kendall's $\tau$ is a more robust ordinal metric, less sensitive to ties and outliers.

Figure 7a shows that the original memorization scores and the Feldman estimator are strongly correlated (Pearson $r = 0.76$, Spearman $\rho = 0.71$, and Kendall $\tau = 0.50$), whereas the other proxies exhibit much weaker associations. Among the distance-based metrics, cosine distance shows the strongest correlation with memorization scores ($r = 0.38$), although the obtained value suggest that the corelation is weak. The Mahalanobis distance variants show even weaker correlations ($r = 0.31$ for normalized, $r = 0.28$ for standard), indicating that statistical distance from class distributions has very limited predictive power for memorization. Notably, Euclidean distance ($r = 0.08$) and iCaRL ranks ($r = 0.02$), while commonly used for exemplar selection, show almost no correlation with memorization scores, suggesting that simple geometric proximity to the class mean is largely orthogonal to memorization patterns. The negative correlations observed for Carlini-Wagner distance ($r = -0.15$) and LASS distance ($r = -0.15$) suggest these adversarial robustness measures may actually be inversely related to memorization, possibly because memorized samples are more vulnerable to adversarial perturbations and this vulnerability may capture different properties than those reflected in memorization scores. Overall, these results highlight that most standard feature-space distances are poor proxies for true memorization, with the Feldman estimator remaining the most reliable among those tested.

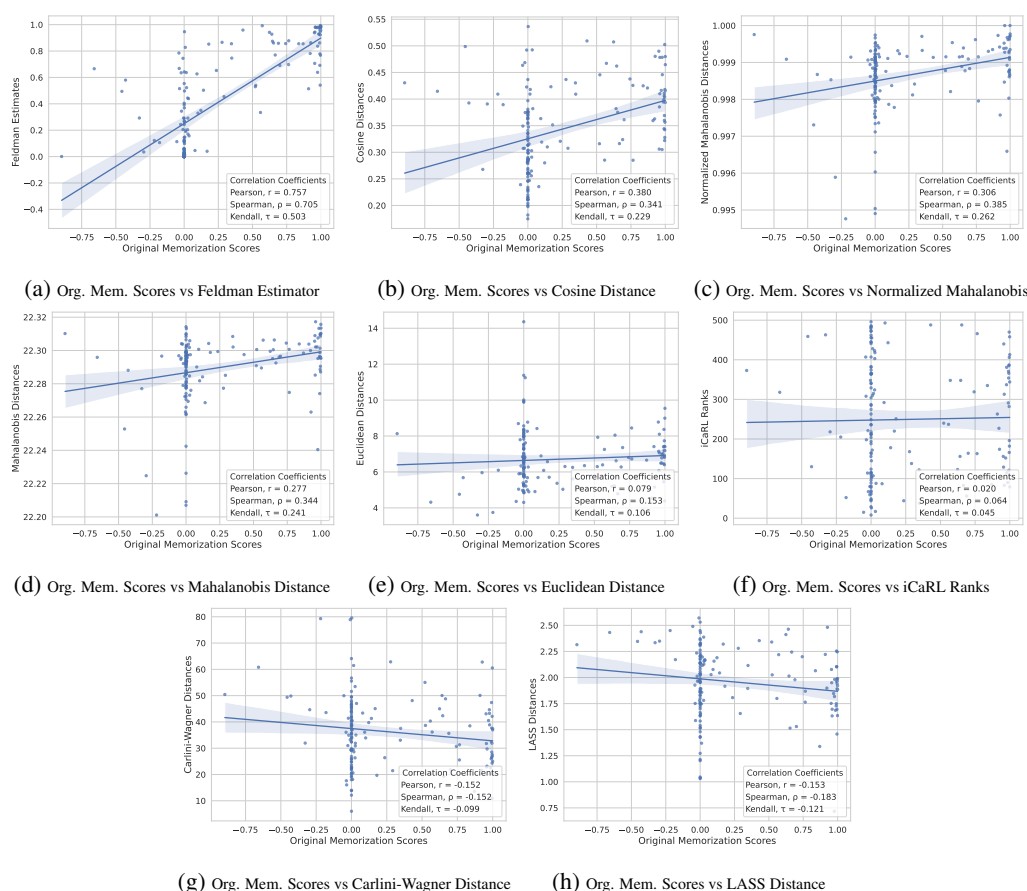

(a) Org. Mem. Scores vs Feldman Estimator     (b) Org. Mem. Scores vs Cosine Distance     (c) Org. Mem. Scores vs Normalized Mahalanobis

(d) Org. Mem. Scores vs Mahalanobis Distance     (e) Org. Mem. Scores vs Euclidean Distance     (f) Org. Mem. Scores vs iCaRL Ranks

(g) Org. Mem. Scores vs Carlini-Wagner Distance     (h) Org. Mem. Scores vs LASS Distance

Figure 7: Correlation between original memorization scores and various proxies

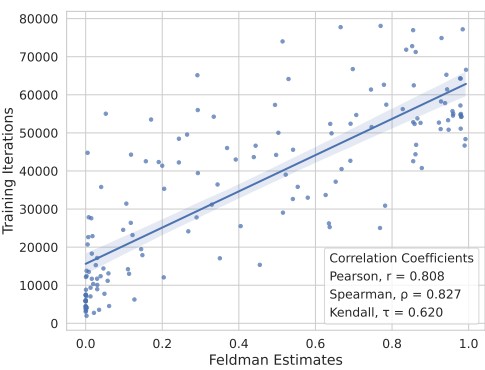

Figure 8: Feldman Estimator vs Training Iteration Estimates

Following the findings from Fig. 7a, we further demonstrate that our training-iteration proxy follows the same pattern as the Feldman estimator by comparing them directly in Fig. 8, again observing strong agreement. Our training-iteration proxy achieves Pearson $r = 0.81$, Spearman $\rho = 0.83$, and Kendall $\tau = 0.62$ against the Feldman estimator, indicating it is a reasonable approximation. Since we have established that the Feldman estimator tightly tracks the original memorization scores, and our proxy tightly tracks the Feldman estimator, we can confidently rely on the training-iteration proxy as a computationally efficient alternative.

Since we use a single training run to compute original memorization scores, our estimate could be biased. However, given the moderate to high correlations with both the original memorization scores and the Feldman estimator, we can deduce that samples learned later in training can effectively identify highly memorized learning examples.

## D  TRAINING HYPERPARAMETERS

We use a standard set of hyperparameters that are commonly used for various Continual Learning algorithms (Buzzega et al., 2020a; Boschini et al., 2023). For both the continual and stationary cases, we train with SGD for 50 epochs per task, or in the case of non-incremental training, we train for 50 epochs on the entire dataset. All networks (including baselines) are trained with a learning rate of 0.1, which is divided by 10 at epochs 35 and 45. In case of non-incremental training, we used a weight decay of 1e-06 and momentum equal to 0.9. For continual training, we set both weight decay and momentum to 0.0. For all training setups, we use a batch size of 32 for both the current data and samples from the memory buffer. For all experiments with continual learning benchmarks, we provide on our repository a complete registry of our experiment results in MLFlow (Chen et al., 2020) along with hyperparameters used to start each training run. We store each commandline argument used for training as an MLFlow hyperparameter, therefore our results could be easily inspected. We did not keep such a registry for our experiments with the memorization score.

The only exception is training LwF, where we used alpha (regularization hyperparameter of knowledge distillation loss) equal to 0.99, learning rate 0.02, 10 epochs, SGD momentum of 0.0, and weight decay 5e-4. We found that this set of hyperparameters works better than the standard set of hyperparameters used for other methods.

For all datasets in experiments, we use an augmentation pipeline consisting of the following transformations:

1. Random Horizontal Flip with probability 0.5

2. Random Crop with size of the original image (32 for CIFAR datasets, and 64 for TinyImageNet) and padding of 4

3. Transformation to tensor - that normalizes image pixels into [0,1] interval

4. Normalization with channel-wise mean and standard deviation computed for each dataset separately.

This is a standard set of augmentations that is available in Torchvision library.

## E  ADDITIONAL MEMORIZATION SCORE RESULTS

**CIFAR100 subsets**   We provide the memorization plots for subsets of CIFAR100 with different size in Fig. 9. As different subsets has different number of samples, the left-hand side of the plot has largest bars for whole dataset. However, the mean and number of samples plots reveal, that memorization increases as the dataset subset get smaller. The only exception in our experiments for that is the smallest subset of size 0.1. To examine this case we plot the memorization for 0.1 subset of CIFAR100 in Fig. 10. We see that the mode of the memorization scores distribution has shifted to the right, indicating general increase in the memorization values. This phenomena may arise due to low number of training samples, where network cannot learn well-generalizable features and must resort to the memorization.

**CIFAR10 subsets**   For completeness, we also provide the histogram of memorization scores for different CIFAR10 subsets. The results are presented in 11. Similarly as in the case of CIFAR100 the memorization increases as the dataset size subset get smaller, with exception of subset size 0.1. In Fig. 12 we present the histograms for different subset sizes on separate plots. For a subset of size 0.1, we can see that the whole distribution of memorization scores shifted its mode from around zero to 0.1. This caused the skew in the mean value and fraction of memorization scores above 0.25. Similarly to case of CIFAR100 this could be caused by too small dataset size, and increased need for memorization. We did not discuss this phenomenon in the main part of our paper due to the limitation of page size.

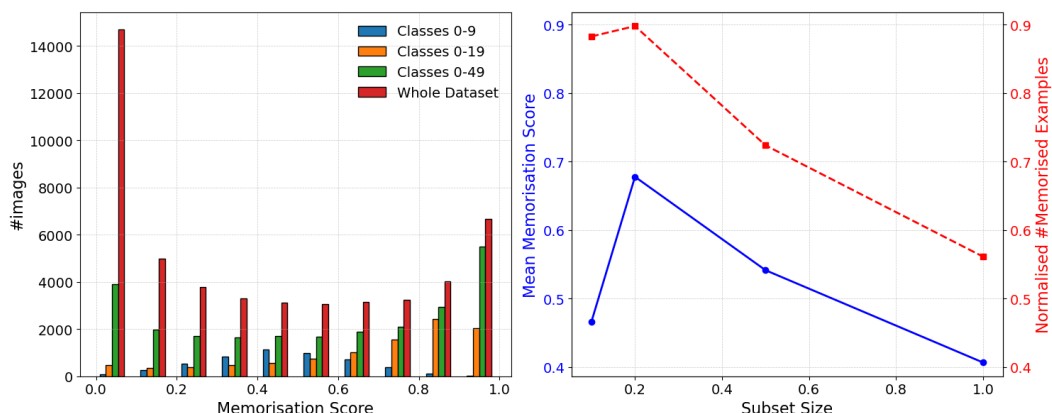

Figure 9: (Left) histogram of proxy memorization scores for different CIFAR100 subsets. (Right) mean memorization score and number of memorized samples normalized by subset size for the data from left plot.

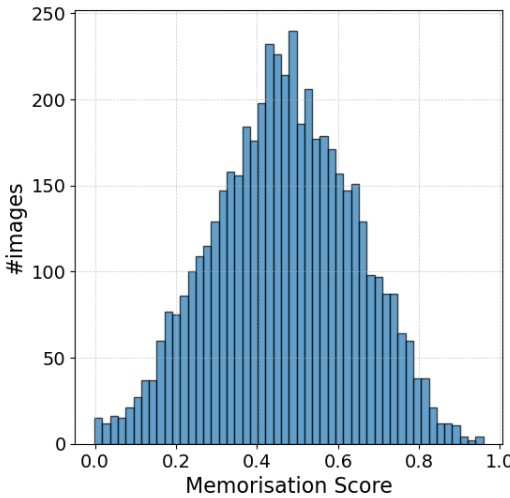

Figure 10: Histogram of memorization scores for the networks trained on 0.1 subset of CIFAR100.

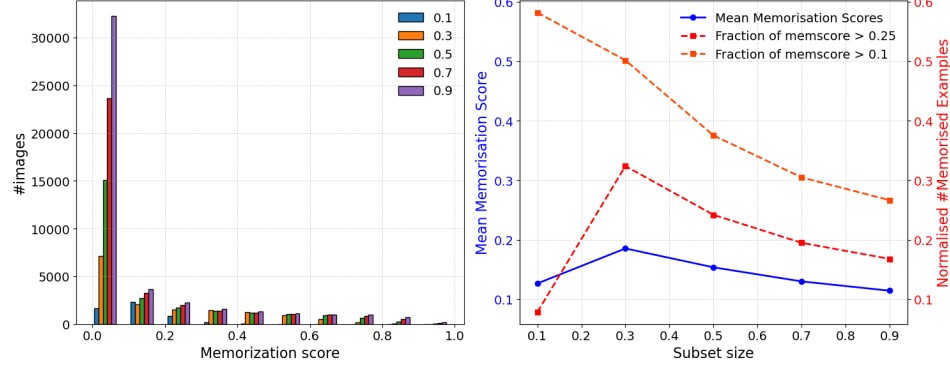

Figure 11: (Left) histogram of proxy memorization scores for different CIFAR10 subsets. (Right) mean memorization score and number of memorized samples normalized by subset size for the data from left plot.

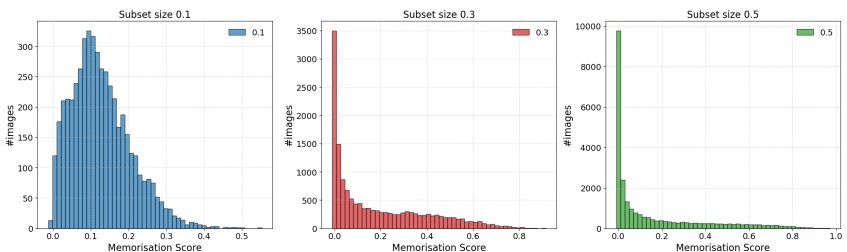

Figure 12: Histograms of memorization scores for different CIFAR10 subsets

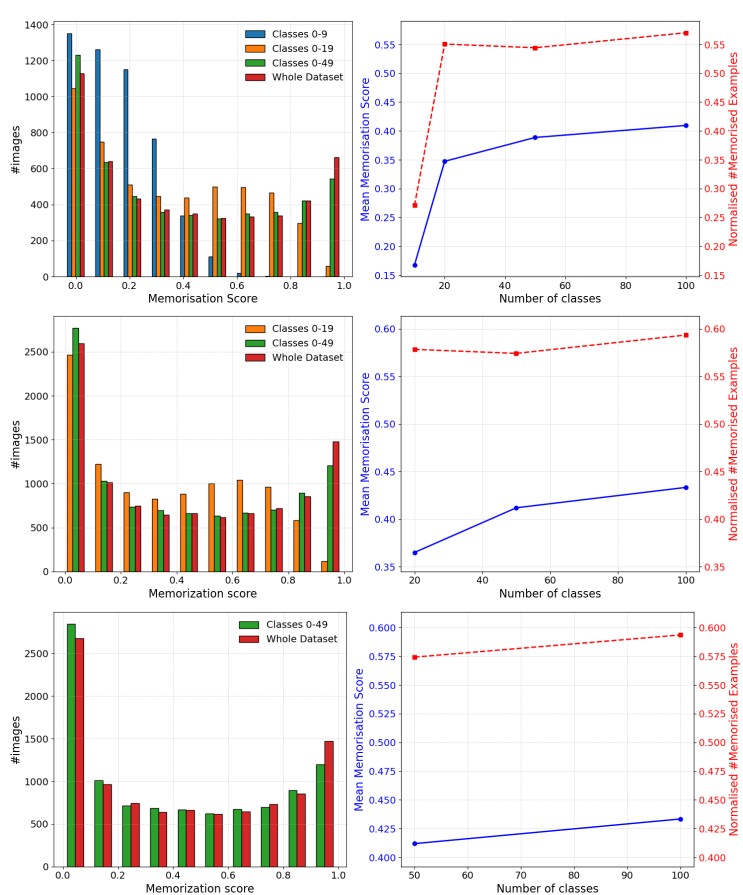

Figure 13: Histograms of memorization scores (left) and mean memorization score with normalized number of examples (right) for the first 10 (top), 20 (middle), and 50 (bottom) classes.

**CIFAR100 other class sets**    The main part of our paper has memorization scores computed for the first 10 classes of CIFAR100 for different class subsets used for training. Here we provide plots for the first 20 and 50 classes as well in Fig. 13.

In our earlier experiment description, we made the comment that the influence of the larger number of classes outweighs the influence of the smaller number of samples, without justifying it. With exact results for both CIFAR100 and CIFAR10 in this appendix, we can now justify our statement. For 50 classes trained on CIFAR100 (Fig. 13 bottom), the mean memorization score is above 0.4. For CIFAR10 with 0.5 samples (see Fig. 11 right), the mean memorization score is below 0.2. These networks were trained with an equal number of samples, but with different numbers of classes, showing that indeed the number of classes has a stronger influence than the number of samples.

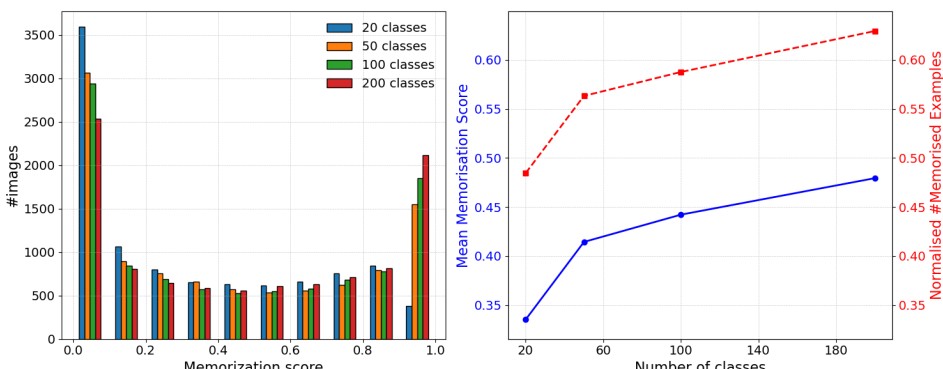

Figure 14: (Left) Histogram of proxy memorization scores for TinyImageNet dataset with various numbers of classes. (Right) Mean memorization scores and fraction of samples from dataset with memorization score above 0.25.

## F    MEMORIZATION SCORES FOR TINY IMAGENET

We test if our observations of the relation between the number of classes and memorization are robust and occur also in another dataset. We test Tiny ImageNet (Wu, 2017), as it has a larger image resolution and a larger number of classes. We present our results in Fig. 14. We notice a very similar trend to the one observed in the main part of the paper - both the average memorization score and fraction of samples with memorization scores above 0.25 rise. This additional experiment results increase credibility of our conclusion, however, more experiments are needed to check if these observations hold for datasets with even larger image resolution, a larger number of images, different hyperparameters, and with different network architectures.

## G    IMPACT OF WEIGHT DECAY ON MEMORIZATION SCORES

In Continual Learning, the networks are often trained with small or no weight decay regularization, especially in case of rehearsal-based methods Buzzega et al. (2020a); Boschini et al. (2023); Harun et al. (2024a). In our experiments with deeper architectures we needed to use larger weight decay regularization, to enable training on the same datasets but with larger capacity. To establish what is the impact of the L2 regularization on memorization we trained several ResNet18 networks on CIFAR100 with different values of weight decay. The results are presented in Fig. 15. We found, that training with low weight decay or no weight decay regularization increases the memorization scores of the networks. It means that low regularization coefficients used in incremental learning promote more memorization, leading to increased forgetting rate, as demonstrated in prior experiments. These results suggest, that we should rethink the usage of regularization in memory-based incremental training.

## H    INCREMENTAL REPRESENTATION LEARNING

Hess et al. (Hess et al., 2024) studied forgetting in representations using linear probes (LPs), demonstrating that forgetting should be evaluated in relation to performance before and after training on a given task. In this work, we replicate their experiment, incorporating the memorization score as an additional factor. Specifically, we train the linear probes on data from task 5, using frozen representations obtained from a ResNet18 model trained incrementally with SGD on Seq-Cifar100. We use hyperparameters reported in the original article, including the AdamW (Loshchilov & Hutter, 2019) optimizer, a learning rate of 0.001, weight decay of 0.0005, and a batch size of 128. As shown in Fig. 16, our results mirror the findings of the original the paper, with the LP accuracy rapidly dropping back to the values before training after exposure to a new task. To investigate the relationship between features and memorization, we partition the test set into subsets according to memorization score. Two key observations arise from this analysis. First, irrespective of memorization score, linear

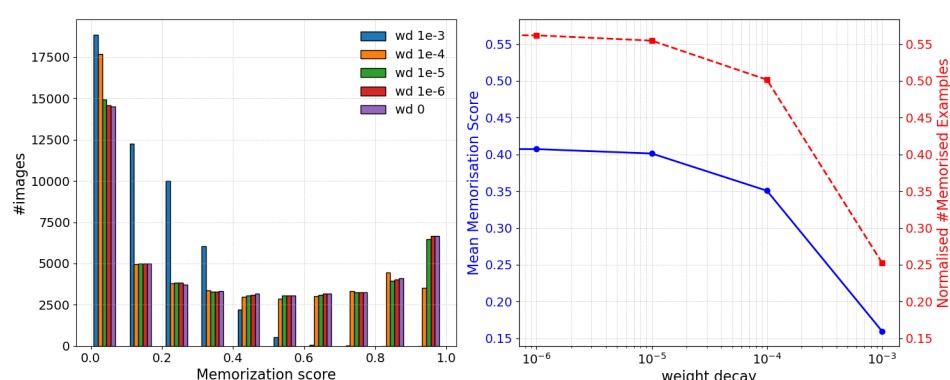

Figure 15: (Left) Histogram of proxy memorization scores for ResNet18 trained on CIFAR100 dataset with different values of weight decay. (Right) Mean memorization scores and fraction of samples from dataset with memorization score above 0.25.

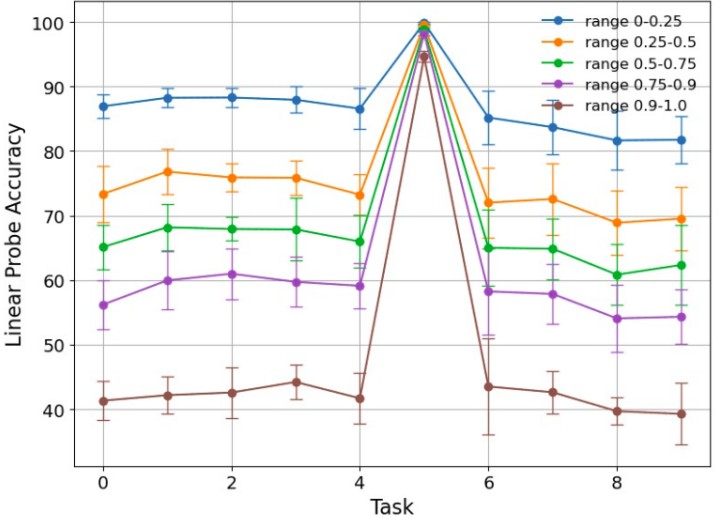

Figure 16: Linear probe accuracy for task 5 during incremental training with SGD on Seq-Cifar100. Results averaged over 5 runs.

probe accuracy exhibits a consistent trajectory: it peaks during the training task and subsequently declines to its pretraining level. Second, samples with higher memorization scores have weaker linear probe performance, whereas samples with lower memorization scores attain higher accuracy. We conjecture that low memorization samples encode more transferable, task-agnostic features that are effectively captured by the feature extractor even when trained on different tasks.

Fig. 16 shows the same dynamic for training and test data, even with the highest memorization threshold.

Previous work on memorization in stationary training (Feldman & Zhang, 2020) showed that memorization is present in representation and not in the classifier. In our experiment, LP trained on representations from other tasks can correctly classify a substantial portion of the data, even with the highest memorization. It may suggest that data with high memorization scores computed offline doesn't necessarily correspond to data that is actually memorized in incremental training. It means we must be careful with our analysis, as determining what is remembered during incremental analysis requires a different method.

## I    TEST SET ACCURACY FOR DIFFERENT MEMORIZATION THRESHOLDS

We provide results for memorization thresholds different from those used in the main part of the paper. The value of 0.25 used in the main part of the paper is consistent with prior literature (Feldman & Zhang, 2020), but it is rather arbitrary. For this reason, we provide our results with thresholds for determining memorized samples equal to 0.5, 0.75, and 0.9. The results plotted in Fig. 17 suggest that the general trends we reported in the paper also hold for other memorization score thresholds.

## J    IMPLEMENTATION AND FULL ALGORITHM

We provide the full algorithm for Memorization-aware Experience Replay in Listing 1. We use three selectors in our paper:

- **top-k** - selecting the k samples with highest memorization score proxy. We do not consider samples that were never correctly classified (i.e., we ignore samples with $v_i = \infty$).

- **middle-k** - selecting the k samples with middle memorization score proxy. This requires sorting the proxy memorization scores, to select the median, and then taking k/2 samples with higher and lower samples than median. In the case of memorization score proxy adapted in this paper, the training iterations order does not require sorting, therefore no additional computational cost is needed.

- **bottom-k** - selecting the k samples with lowest memorization score proxy.

In Listing 1, we use mathematical notation in line 8 to indicate obtaining the index of samples in the dataset (at what position in the dataset the given sample is stored). In practice, however, we modify the implementation of the dataset to return the index along the image and label pair, to avoid the expensive search for a matching sample. In line 23, we update the buffer by selecting the most frequent classes and replacing the buffer samples that we randomly draw from the current most frequent class.

## K    TIME COMPLEXITY

We provide the relative execution time for the proposed method and other baselines for all datasets used in experiments in Tab. 4. Many factors can impact the program execution time, such as specific CPU or GPU settings. We did not provide accurate benchmarks that consider these factors. We provide only rough estimates based on our logs. We do not claim that these values are correct execution time benchmarks, and they should not be treated as such. Another factor that could potentially impact the execution time is the implementation. We did not optimize for speed in any of the baselines that we used. This could also heavily impact the results of execution time benchmarks.

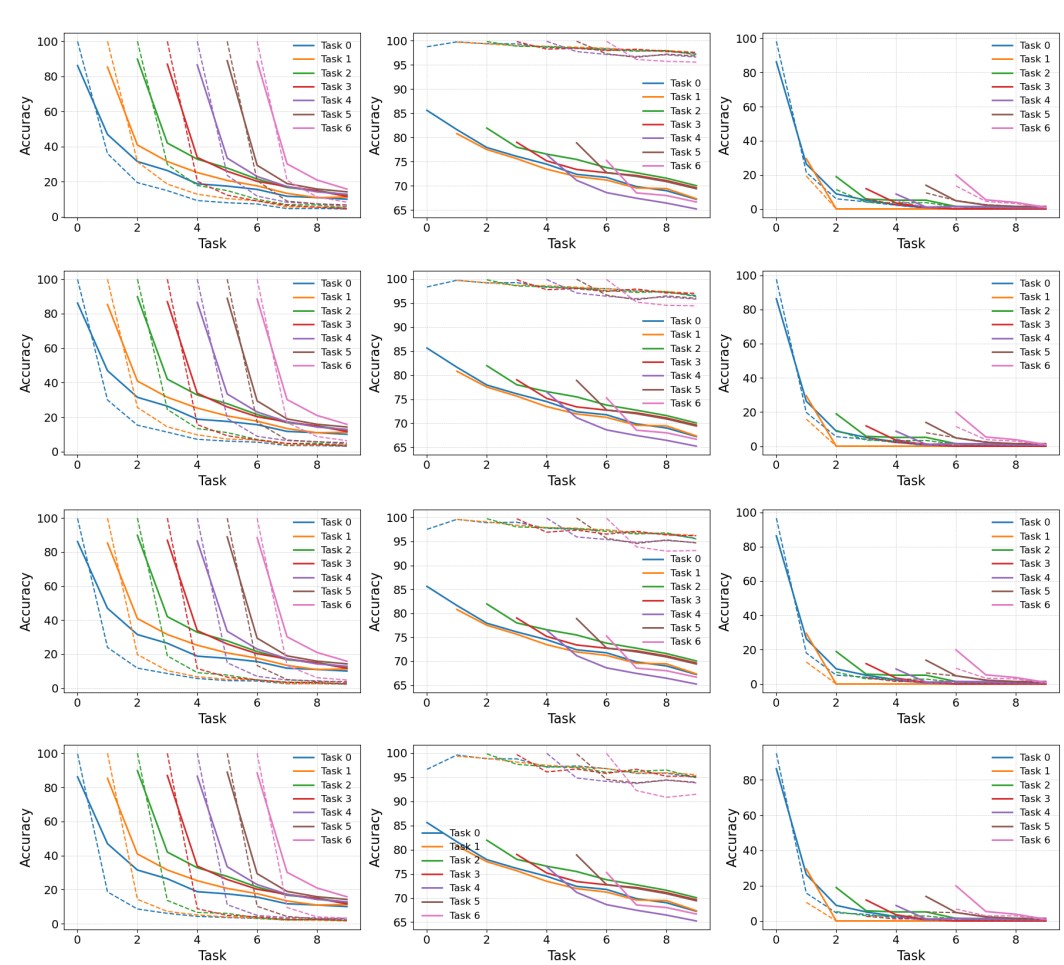

Figure 17: Task accuracy for test set (solid line) and samples with high memorization (dotted line) across incremental training on Seq-Cifar100 stream with 10 tasks. (Left) training with buffer size 500. (Middle) training with full access to previous tasks. (Right) training with LwF. The rows correspond to memorization score thresholds of 0.25, 0.5, 0.75, 0.9. Results averaged over 5 runs.

Table 4: Relative execution time for the proposed method and baselines used in our experiments.

| buffer policy | relative execution time | | |
|---|---|---|---|
| | Split-Cifar10 | Split-Cifar100 | Split-TinyImageNet |
| reservoir | 1.0 | 1.0 | 1.0 |
| reservoir balanced | 0.98 | 1.01 | 1.00 |
| rainbow memory | 1.06 | 1.10 | 1.03 |
| PBCS | 2.05 | 2.43 | 2.00 |
| BCSR | 1.09 | 1.18 | 1.19 |
| bottom-k memscores | 1.01 | 1.06 | 1.00 |
| middle-k memsocre | 1.01 | 1.06 | 1.00 |
| top-k memscores | 1.03 | 1.10 | 1.03 |

**Algorithm 1** Memorization-aware Experience Replay

**Require:** $S = \{D_1, D_2, ...\}$ - stream with tasks, $f(\theta)$ - network, $\mathcal{M}$ - memory buffer, $Q$ - selector (top-k, or other)

1:   $t \leftarrow 0$
2:   **while** $D_t$ arrives **do**
3:      $v \leftarrow [\infty, \infty, ..., \infty]^T \in \mathbb{R}^{n_t}$
4:      $iter \leftarrow 0$
5:      **for** $X, Y \sim D_t$ **do**
6:        $\hat{Y} \leftarrow f(X, \theta)$
7:        **for** $x, y, \hat{y} \in X, Y, \hat{Y}$ **do**
8:          $i \leftarrow D_{index}[(x, y)]$
9:          **if** $\hat{y} = y$ AND $v_i = \infty$ **then**
10:           $v_i \leftarrow iter$
11:          **end if**
12:          **if** $\hat{y} \neq y$ AND $v_i \neq \infty$ **then**
13:           $v_i \leftarrow \infty$
14:          **end if**
15:        **end for**
16:        $X_m, Y_m \leftarrow \mathcal{M}$
17:        $\mathcal{L} \leftarrow \mathcal{L}(\hat{Y}, Y) + \mathcal{L}(f(X_m, \theta), Y_m)$
18:        $\theta \leftarrow \theta - \lambda \nabla_\theta \mathcal{L}$
19:        *reservoir_sampling* $(\mathcal{M}, X, Y)$
20:        $iter \leftarrow iter + 1$
21:      **end for**
22:      $D_s \leftarrow \emptyset$
23:      **for** $c \in C_t$ **do**
24:        $k \leftarrow \{i : y_i = c, y_i \in D_t\}$
25:        $v_s \leftarrow Q(v[k], |\mathcal{M}|/(t+1))$
26:        $D_s \leftarrow D_s \cup \{(x_j, y_j) \in D_t | j \in v_s\}$
27:      **end for**
28:      update $\mathcal{M}$ with $D_s$
29: **end while**

## L  COMPUTATIONAL RESOURCES

We used three machines to perform computations in this work:

- machine with: 1x NVIDIA RTX3090 and 128Gb RAM memory
- machine with 2x NVIDIA RTX3090 and 64Gb RAM memory
- server with 8x NVIDIA A5000 and 128 Gb RAM memory

Training a single ResNet18 on full Cifar100 required approximately 10 minutes for our implementation. We used the standard implementation of Pytorch training, as we needed flexibility in terms of creating subsets of the data in a controlled manner. We provide the GPU hours required to reproduce our experiments:

- impact of number of classes on memorization Cifar100: 82 hours
- impact of dataset size on memorization: 113 hours
- impact of depth on memorization: 185 hours (only ResNet34 and ResNet50, as ResNet18 was part of the previous experiment)
- impact of width on memorization: 147 hours (only widths 0.25, 0.5 and 0.75, full width was part of previous experiment)
- impact of number of classes on memorization TinyImageNet: 290 hours
- memorization in incremental training: 10 hours
- incremental representation learning: 5 hours
- memorization score proxy: 51 hours
- evaluation with standard benchmarks: 241 hours (only the proposed methods, not the baselines)
- training with larger buffers: 148 hours (only the proposed methods, not the baselines)

We round up the numbers above to a full hour. Some of the experiments were run on both RTX3090 and A5000 GPUs, therefore we do not differentiate between compute time on these different cards. The compute time should be valid for graphic cards with both compute capability and VRAM equal to or larger than ones provided by RTX3090. We do not provide the compute required for some of the experiments in the appendices, as it would be hard to obtain exact values. We estimate the full duration of the experiments of our whole project to be 1476 hours. Full research project required larger compute resources than one stated above for the running of experiments, as we did several preliminary experiments, and some runs failed due to errors in code or configuration.

