# OpenReview forum: "What is the role of memorization in Continual Learning?"
_ICLR.cc/2026/Conference — Submitted to ICLR 2026_

### Official Review · Reviewer_1Aok · 2025-10-16

**Soundness:** 2
**Presentation:** 3
**Contribution:** 2
**Rating:** 4
**Confidence:** 4

**Summary:**

The paper investigates the role of memorization in continual learning. The authors argue that memorization can both enhance performance and accelerate forgetting. The main contribution of the paper lies in proposing a memorization proxy and validating its usefulness in buffer policy design.

**Strengths:**

1. The paper explores the relationship between memorization and forgetting in continual learning.
2. Since existing evaluation methods for memorization scores are not directly applicable to continual learning, the authors propose a memorization score proxy as an approximate substitute.
3. For memory-based continual learning methods, the proposed approach provides insights for guiding example selection.

**Weaknesses:**

1. Most of the paper’s conclusions are empirically derived, lacking theoretical support for the relationship between memorization and forgetting in continual learning.
2. The proposed proxy in Eq. 3 assumes a positive correlation with memorization degree. This assumption lacks theoretical justification, and the reported correlation coefficient indicates only a moderate relationship. Such a weak correlation could introduce noise into buffer example selection, making the results difficult to interpret.
3. One of the paper’s core contributions is the memorization-aware replay method; however, the experimental evidence does not strongly support the claim that “storing high-memorization samples is beneficial.” In fact, the top-k strategy performs significantly worse than bottom-k and mid-k in nearly all benchmarks, suggesting that storing high-memorization samples is detrimental when the buffer size is small. Moreover, mixing 10% of top-k samples into the bottom-k or mid-k buffers leads to negligible (or even negative) gains, all within the margin of standard deviation. Thus, the conclusion that “the importance of high-memorization samples increases with larger buffer size” seems weakly supported by the data.
4. To convincingly demonstrate that “memorization is necessary to achieve the highest performance,” an experiment capable of actively suppressing memorization without affecting pattern learning would be required. However, such isolation of the memorization variable is extremely challenging, and the current experiments cannot achieve this level of control.
5. The paper defines memorization at the sample level, yet its discussion of forgetting seems to operate at the representation level. For instance, is a high-memorization sample forgotten because it relies on fragile or rare features? If the model memorizes many such samples, might it gradually learn more generalizable representations to capture them, thereby improving overall performance? This conceptual ambiguity leaves it unclear whether memorization should ultimately be suppressed or encouraged.

**Questions:**

Please refer to Weaknees.

---

> ### Author Response · Authors · 2025-12-03
>
> We appreciate thorough analysis and valuable comments from the Reviewer. Below are our answers to the 5 weaknesses pointed out by the Reviewer:
>
> 1. We agree that our study is primarily empirical in nature. Our goal is to systematically characterize how memorization correlates with forgetting in realistic CL settings, not to derive a full theory. In the revision we will (i) explicitly frame the contribution as an empirical investigation, (ii) connect our observations to existing theoretical work on memorization and long-tail effects, and (iii) add a dedicated Limitations paragraph highlighting the lack of formal analysis as an open direction.
>
> 2. Our proxy does not assume a strong linear relationship with the Feldman score; it is used as a cheap ranking heuristic rather than a calibrated estimator. Moderate rank correlations (Pearson/Spearman ≈0.6) are sufficient for this purpose and are empirically validated by the fact that bottom-/mid-k buffers consistently outperform reservoir sampling. We will clarify this in Sec. 3.4/3.5, explicitly present the proxy as a heuristic, and temper claims about its precision.
>
> 3. We agree that high-memorization samples are not beneficial in the small-buffer regime, and we will soften our wording accordingly. Our intended claim is that their relative importance grows as memory approaches the “large buffer” regime, where top-k performance moves closer to the full-memory upper bound and small positive effects appear when mixing 10% top-k. We will (i) emphasize the negative results for small buffers, (ii) rephrase the conclusion to highlight a qualitative trend rather than a strong statement, and (iii) add a short discussion on the limited effect sizes in Tab. 3.
>
> 4. We fully agree that cleanly isolating memorization—while holding pattern learning fixed—would require an idealized intervention that is beyond the scope of our current setup. Our statement that “memorization is necessary for highest performance” is meant as an empirical corroboration of Feldman’s observations under CL training, not as a causal proof. In the revision we will soften this claim, clearly label it as observational, and explicitly discuss the impossibility of perfectly controlling for memorization in realistic experiments.
>
> 5. Thank you for pointing out this conceptual mismatch. In our experiments, forgetting is still quantified at the sample level (accuracy on the same examples over time); references to “representation-level” effects were meant as an interpretation of why certain samples are forgotten faster (e.g., reliance on rare features). We will clarify this by (i) explicitly distinguishing sample- and representation-level notions, (ii) adding a short discussion of possible mechanisms (fragile vs. generalizable features), and (iii) reframing our take-away as: CL should neither blindly suppress nor encourage memorization, but understand when each regime is helpful.

---

### Official Review · Reviewer_s1KA · 2025-10-29

**Soundness:** 2
**Presentation:** 3
**Contribution:** 2
**Rating:** 4
**Confidence:** 3

**Summary:**

This paper investigates the role of memorization in the context of continual learning (CL). The authors define a memorization score for individual examples (based on prior definitions from Feldman et al.), and study how examples with high memorization scores behave under incremental training regimes. The work thus provides empirical insight into how “memorization” and “forgetting” interact in CL, and offers practical suggestions for buffer‐selection policies.

**Strengths:**

1. Through well-designed experiments on standard continual learning benchmarks (such as the CIFAR100 split), the authors reveal a pronounced correlation between memory scores, buffer size, and forgetting behavior.
2. The explicit study of how memorized examples behave under continual learning is a valuable contribution.
3. I think the paper is well-written and generally easy to follow.

**Weaknesses:**

1. Despite robust experimental validation, the paper offers a limited theoretical explanation for why highly memorized samples are more prone to forgetting in continual learning.
2. This study argues that memory should be considered in incremental training, but it does not explore how much additional computational or memory overhead would be introduced by calculating memory scores or implementing proxy strategies in practical applications.

**Questions:**

1. Have you tested the proxy memorization buffer policy on non‐vision tasks (e.g., NLP, tabular, reinforcement‐learning streams) to assess generality?
2. Can you provide insight into why high‐memorization examples are more quickly forgotten in continual sequences? Is it because they occupy more capacity, are less “representative”, or because the model shifts representation more aggressively?
3. In continual learning, retaining generalizable features is often more important than memorizing individual examples. Can you explain the differences and connections between these two?

---

> ### Author Response · Authors · 2025-12-03
>
> We appreciate the time and effort of the Reviewer spent on evaluating our submission. Below we present answers to the questions raised by the Reviewer:
>
> 1. Thank you for this question, it is certainly an interesting research area. Our work focused on the class-incremental learning scenario, and the choice of data type is primarily dedicated by the fact that this type of data in the field of visual pattern recognition is often used to evaluate models. However, the phenomenon of memorization is not unique to vision-related tasks, but also applies to LLM, for example, which has been identified as one of the important privacy issues associated with this type of model (see, for example, https://arxiv.org/abs/2202.07646). However, in the case of LLM, it would be difficult for us to propose an experiment for the class-incremental scenario. Other types of data, including tabular data, are less susceptible to the memorization effect and are often well generalized, but we did not actually consider them in our experiments.
>
> 2. In our setting, memorization follows Feldman’s definition: a sample has a high memorization score if its correct classification strongly depends on its presence in the training set, which typically happens for atypical, long-tail, or weakly represented patterns. In other words, high-memorization examples are precisely those that cannot be well supported by the shared, “typical’’ features learned for the rest of the class.
>
>  Our experiments indicate that these samples are indeed more fragile under incremental updates. When we track accuracy separately for different memorization strata on Split-CIFAR100, the accuracy of the high-memorization subset decays roughly twice as fast as that of low-memorization points (Fig. 1-right and Fig. 4-right). This happens even though, in the stationary setting with full access to all data, memorized samples maintain high accuracy across training, suggesting that they are not inherently “unlearnable’’ but rather particularly vulnerable to distribution shifts and lack of rehearsal.
>
>  Our interpretation, consistent with these results, is that high-memorization examples occupy narrow, idiosyncratic regions of representation space. They rely on late-stage, task-specific adjustments (as also observed by Arpit et al.) and have little support from nearby samples. During continual training, gradients are dominated by abundant, typical examples from new tasks; the representation is reorganized toward more generic features, and the small regions devoted to idiosyncratic/long-tail examples are the first to be overwritten. This is reinforced by capacity constraints: as we increase the number of classes with a fixed backbone, the fraction of data that must be memorized rises, so the model has to “reallocate’’ capacity away from earlier memorized tails when new tasks arrive.
>
>  Our hyperparameter and architecture study supports this view. Wider models and weaker regularization increase memorization but also correlate with stronger plasticity and larger forgetting, whereas deeper models and stronger weight decay reduce memorization yet forget more in the continual setting. Together, these observations suggest that high-memorization examples are forgotten faster not because they occupy more capacity per se, but because they are (i) less representative, (ii) supported by non-transferable, late-learned features, and (iii) therefore disproportionately affected by representation drift when the model is adapted to new tasks under limited rehearsal.

---

> > ### Author Response · Authors · 2025-12-03
> >
> > 3. We agree with the reviewer that retaining generalizable features is central in continual learning, and our paper explicitly separates this notion from sample-level memorization. Following Feldman, a high memorization score means that removing a single training point significantly changes its prediction; this is a per-example notion. In contrast, “generalizable features’’ refer to shared representations that support many samples and often multiple tasks. In continual learning, knowledge retention aims at preserving these shared features so that performance on earlier tasks remains high, regardless of whether specific tail examples are still remembered.
> >
> > Our experiments indicate that these two concepts are related but not identical. First, we show that some degree of memorization is necessary to match the performance of stationary training: when we train with full access to past data, memorized samples maintain high accuracy throughout the sequence, and increasing the number of classes (with fixed capacity) systematically increases the fraction of data that must be memorized. This aligns with the long-tail view that rare, atypical test points can only be covered via memorization.
> >
> > At the same time, our linear-probe analysis suggests that low-memorization examples are precisely those supported by more transferable, task-agnostic features. When we train probes on frozen representations and group test points by memorization score, low-memorization subsets achieve higher probe accuracy and maintain similar dynamics across tasks, while high-memorization subsets show weaker linear-probe performance. We conjecture that low-memorization samples lie in well-supported regions of feature space and are thus better captured by generic features that survive across tasks, whereas high-memorization samples rely on more specialized directions that are easier to disrupt.
> >
> > This distinction is also reflected in our buffer-policy experiments. With small memory budgets (buffer size 500), prioritizing low- and medium-memorization examples in the replay buffer yields higher final accuracy and lower forgetting than storing only high-memorization ones, indicating that, in this regime, preserving generalizable patterns is more beneficial than preserving the tails.
> >
> > As the buffer grows, however, policies that include a fraction of high-memorization points improve and approach the full-memory upper bound, suggesting that memorization becomes increasingly important once we have already secured the core generalizable features.
> >
> > In summary, our view—supported by the experiments—is that generalizable features and memorization play complementary roles: generalizable features underpin stability across tasks, while memorization is required to recover performance on atypical or long-tail examples. Continual learning algorithms must balance these, prioritizing generalizable features under tight memory and progressively incorporating memorized tails as more capacity or buffer space becomes available.

---

### Official Review · Reviewer_Ho4j · 2025-10-30

**Soundness:** 3
**Presentation:** 3
**Contribution:** 3
**Rating:** 6
**Confidence:** 4

**Summary:**

This paper explores the role of memorization in continual learning (CL), examining how it influences forgetting and overall performance. While both memorization and forgetting prevention aim to retain knowledge, the paper clarifies that memorization refers to learning specific samples—often atypical or rare—whereas CL seeks to preserve generalizable knowledge across sequential tasks. Using Feldman’s definition of memorization score, it shows that samples with higher memorization are forgotten faster, that memorization increases with the number of classes, and that wider networks tend to memorize more while deeper ones memorize less. Since computing exact memorization scores is computationally expensive, the paper proposes a proxy based on the iteration when a sample is first learned and remains correctly classified thereafter, which strongly correlates with Feldman’s estimator. It integrates this proxy into a “memorization-aware experience replay” buffer policy, finding that storing low- or medium-memorization samples improves performance when memory is limited, while high-memorization samples become useful when the buffer is large.  Experiments on Split-CIFAR10, CIFAR100, and Tiny ImageNet confirm that memorization is necessary to reach high accuracy but can also accelerate forgetting after distribution shifts.

**Strengths:**

1. Provides a novel and focused investigation of memorization in continual learning, a topic rarely examined beyond generalization or privacy contexts.

2. Clearly distinguishes memorization from forgetting prevention, offering conceptual clarity that helps refine the theoretical framing of continual learning.

3. Includes comprehensive experiments across multiple datasets (CIFAR10, CIFAR100, Tiny ImageNet), architectures (ResNet18/34/50), and memory settings, ensuring robust and generalizable findings.

4. Introduces a computationally efficient memorization proxy that correlates strongly with Feldman’s estimator, enabling large-scale empirical evaluation.

5. Demonstrates how memorization-aware buffer policies can guide experience replay strategies and improve continual learning performance under varying memory budgets.

6. Presents well-structured and reproducible methodology, with code availability and detailed hyperparameter reporting.

**Weaknesses:**

1. The paper is at times difficult to follow, and several descriptions lack precision. For example, in Figure 5, it is unclear what the training iterations on the y-axis represent. Upon checking the appendix, it appears that the y-axis corresponds to the proposed training-iteration–based memorization proxy, but this is not clearly stated in the main text. If this interpretation is incorrect, please clarify what is actually plotted. The manuscript would benefit from improved figure captions and clearer explanations of the plotted variables.

2. While the paper aims to study memorization in continual learning, all memorization scores are computed offline using Feldman’s estimator under stationary training and are not updated dynamically during incremental training. The scores are then tracked across tasks as the data distribution shifts. Consequently, the claim—that “memorized samples are forgotten faster”—rests on static pre-training memorization scores, which may not reflect what is actually memorized by the continually updated model. This design limits the causal validity of the conclusion that memorization directly influences forgetting in continual learning.

3. The proposed proxy—defined as the first iteration at which a sample is correctly classified and remains so thereafter—correlates with Feldman’s estimator (r ≈ 0.8), but may actually capture learning dynamics (e.g., sample learning speed or gradient stability) rather than memorization itself. For example, early-learned and consistently correct samples could receive low proxy scores even if they are later forgotten, while atypical or noisy samples that stabilize late may appear highly “memorized.” The paper would benefit from a qualitative analysis (e.g., visualization of high-proxy samples or representation trajectories) to confirm that high proxy values indeed correspond to memorized behavior as per Feldman’s definition—memorization without generalization.

4. Although the empirical findings (e.g., that highly memorized samples are forgotten faster) are intriguing, the paper does not provide a theoretical rationale or mechanistic explanation for these effects. Are the observed patterns due to feature drift, gradient interference, or representational collapse? Furthermore, the claim that “wider models memorize more but forget less” appears contradictory to the earlier finding that memorization accelerates forgetting, yet this paradox remains unresolved. A more formal or analytical treatment would significantly strengthen the contribution.

Minor Comments:

1. Figure 1 (Right Panel): The plot is difficult to read due to small font and contrast. Please improve the figure’s legibility.

2. Section 3.3 (First Line): The equation reference or variable notation is not clearly formatted; please revise for consistency.

3. Reproducibility: Upon acceptance, please consider open-sourcing the trained model checkpoints along with the code, as this would greatly facilitate future research and validation.

4. Table 1: Please include citations for all baseline methods (e.g., Reservoir Sampling, Rainbow Memory, BCSR, PBCS) directly in the table for easier cross-referencing.

**Questions:**

1. Could you please clarify precisely what the y-axis (“training iterations”) in Figure 5 represents? Does it correspond to the proposed training-iteration–based memorization proxy, or to actual training steps? If it is the proxy, please make this explicit in the main text and caption.

2. Since all memorization scores are computed offline using Feldman’s estimator under stationary training, how do you justify using these static scores to analyze forgetting in a dynamic continual-learning setup? Have you attempted to recompute or approximate memorization scores during incremental training to confirm that the same samples remain “memorized” over time?

3. How do you ensure that the proposed proxy (first correct-classification iteration) truly measures memorization rather than training difficulty or convergence speed? Have you examined qualitative examples or visualization of samples with high proxy scores to confirm that they exhibit the behavior expected of memorized examples?

4. To the best of my understanding, the correlation between the proxy and Feldman’s estimator (r ≈ 0.8) was computed on a limited subset of CIFAR-100 samples. Could you provide per-class statistics to assess the robustness of this correlation across the full dataset?

5. Can you elaborate on the underlying mechanism explaining why high-memorization samples are forgotten faster? Have you explored whether this effect is driven by feature drift, gradient interference, or differences in representational overlap between tasks?

6. The paper reports that wider models memorize more but forget less, which appears inconsistent with the claim that memorization accelerates forgetting. Could you clarify how these two findings coexist, or provide additional analysis (e.g., gradient alignment or curvature metrics) to reconcile them?

7. When reporting accuracy for “memorized samples,” are these measurements made on the training set or a held-out validation split?

8. Will you release the trained model checkpoints upon acceptance? Doing so would help the community reproduce and extend your findings.

9. Could you improve the readability of Figure 1 (right panel) and revise Section 3.3’s notation for clarity? Also, please include citations for all baselines (Reservoir, Rainbow Memory, BCSR, PBCS) directly in Table 1 for easier cross-reference.

---

### Official Review · Reviewer_Pd4G · 2025-11-02

**Soundness:** 2
**Presentation:** 2
**Contribution:** 2
**Rating:** 2
**Confidence:** 4

**Summary:**

The paper investigates the empirical relationship between sample-level memorization and catastrophic forgetting in class-incremental continual learning (CL). The well-known memorization score proposed by Feldman is used for the study. The central claims of the paper are: (1) Samples with high memorization scores, computed in a stationary (full-dataset) setting, are forgotten more rapidly during incremental training than their low-memorization counterparts. (2) Memorization increases as the number of classes in a dataset grows, as model width grows, and decreases with model depth. (3)  In Continual Learning, the trend is reversed, namely, deeper models forget more while wider models forget less. (4) The experiments suggest that for small buffers, prioritizing samples with low or medium proxy scores is beneficial, while the utility of high-memorization samples increases with larger buffer sizes.

**Strengths:**

* This works is one of the few existing works studying memorization in continual learning, which I found extremely interesting and important, and potentially significant. In this sense, the work would be original and significant. However, in my view the paper suffers from fundamental flaws that invalidate its primary conclusions (see weaknesses).

**Weaknesses:**

* While the paper's title suggests a board study of memorization role in continual learning, the focus is only on class-incremental continual learning, so the stated findings at best could only be applicable to this narrow problem. Thus, the study conducted in Section 3.2 on the role of varying number of classes could be complete irrelevant to board continual learning setups.

* In a similar vein, the paper calculates memorization scores in an offline, stationary setting (i.e., training on the full CIFAR100 dataset) and then using these static scores to analyze sample behavior during dynamic incremental training. This is a significant methodological compromise. The paper itself concedes this flaw in Appendix H, stating "It may suggest that data with high memorization scores computed offline doesn't necessarily correspond to data that is actually memorized in incremental training". A proper study would investigate the process of memorization within CL, which would be the more fundamental question.

* The study in section 3.2 is fundamentally flawed in the following sense: Note that the memorization score definition in eq 1 and eq 2 are model and data distribution dependent. That is, for a given architecture and given data distribution they assess the memorization of different samples. Thus, one cannot use these equations to compare the memorization of different models and/or different data distributions. When changing the model size (depth or width) or the number of classes, we are changing the architecture and data distribution. Thus, I find the majority of the results in the paper, particularly those from section 3.2 invalid.

* The central finding that "memorized samples are forgotten at significantly faster rates" and the stated conundrum in section 6 may be an artifact of the "regularization-free" setting used for continual learning, i.e. using SGD with momentum set to 0.0 and, most critically, weight decay set to 0.0 vs a regularized setting used for offline/ non-continual learning setting. This regularization-free setting is known to promote memorization, rather than an intrinsic property of continual learning. I further encourage the authors to look into the literature on plasticity loss and approaches akin to weight decay/L2-regularization.

* The paper uses eq 3 as a proxy of memorization score. This approach basically uses the first iteration $j$ after which the sample $i$ is always classified correctly. While intuitive, I find its implementation, described in Algorithm 1 flawed. The paper states that checking every sample at every iteration is too costly. The algorithm, therefore, only updates the status of a sample $i$ when it appears in the current minibatch. This transforms the proxy from a deterministic function of the training trajectory into a highly stochastic one, dependent on the minibatch sampling order. A sample could be "learned" at iteration $j=100$ but, by chance, not be sampled again until $j=5000$, at which point its proxy score $v_i$ would be recorded as 5000, not 100 (see line 10 of Algorithm 1). The impact of this substantial noise and bias, which fundamentally separates the implementation (Algorithm 1) from the definition (Eq. 3), is not analyzed.

* Figure 5 shows only a weak, "moderate" correlation (Pearson $r=0.594$) between the proxy and the "Original Memorization Score" stated in eq 1. Note the indirect connection presented in Figure 8 (strong correlation ($r=0.808$) between eq 2 and eq 3) and Figure 7a (strong correlation ($r=0.757$) between eq 1 and eq 2) in the Appendix are not adequate to substantiate the validity of eq 3.

* The performances reported in Tables are too low and have a huge variance to support the conclusions and findings. For instance, the statement "we can see that selecting lower and mid proxy memorization scores obtains better results than reservoir sampling" is not substantiated in my view in light of the results reported in the tables.

**Questions:**

(1) The paper claims to use $k/n=0.5$ and $u=250$ (number of networks trained to approximate the expectation). This implies that for each sample $i$, $2 \times u = 500$ networks must be trained. For the CIFAR100 dataset ($n=50000$), computing this score for all samples would require $500 \times 50000 = 25,000,000$ network-trainings! The paper reports a training time of 10 minutes per network, implying a total compute of approximately 4.16 million GPU-hours. This is irreconcilable with the paper's reported computational budget of "over 3500 neural networks" or the total project time of 1476 hours.

---

> ### Author Response · Authors · 2025-12-03
> **Response to the Reviewer Pd4G**
>
> We would like to thank the reviewer for their time and effort. We are glad that the reviewer found our work important.
>
> We must acknowledge that the paper title is too broad, and we should change it. We also acknowledge that calculating memorization scores offline is not the perfect solution; however, we found no valid alternatives. As training with each new task changes the weights, the initialization and representation space change across the whole incremental training. The order of tasks could also change. This means that the number of random variables that should be taken into consideration grows significantly if we want to compute the memorization score for each task separately. With our computational budget, such a cost would be prohibitive. Work on novel methods of memorization estimation with changing data distributions is a really challenging and important problem. Using precomputed memorization scores is a compromise we made; however, we do not believe that it renders our results invalid. We acknowledge that true memorization during training with each task can be different from precomputed scores, but we have shown that samples with higher precomputed memorization scores exhibit different behavior during incremental training, and taking into account this information can lead to creating an informed buffer policy that improves the results.
>
> The question about the impact of weight decay in our research is a valid one. We will discuss the impact of weight decay in our work.
>
> We do not analyze the impact of the order of batches on the memorization score proxy, because we are interested mostly in the difference between memorizing some samples or selecting samples with higher memorization. When training for 50 epochs, there are some samples that are learned in the first few epochs, and others that are memorized near the end of training. The order of data sampling does not change the estimator value significantly, considering the memorization dynamics. This is exhibited in Figure 8 in Appendix C.
>
> We do not understand the statement about "huge variances". In fact, the variance values in Tables 1 and 2 are on par with other baselines. The results in Table 1 indicate higher accuracy and lower forgetting measures compared to reservoir sampling. In the case of CIFAR100, the difference between bottom-k is higher than the sum of each method's variance, indicating a clear difference between these methods.
>
> Answering the questions:
>
> (1) In our experiments, we don't have to train a separate network for each learning example. In fact, the whole point of the memorization score estimator proposed by Feldman [1] was to circumvent this issue. In our setup, similarly to the one used by Feldman [1], we use $u = 250$ - meaning that we repeat training 250 times with different subsets. Each subset consists of $k/n =0.5$ of the original training dataset. In this way, we can remove multiple samples from the training data for a single training.
>
> References:
>
> [1] Vitaly Feldman, \& Chiyuan Zhang. (2020). What Neural Networks Memorize and Why: Discovering the Long Tail via Influence Estimation.

---

### Meta-Review · Area_Chair_C1vG · 2026-01-06

**Summary:**

One reviewer raised an important concern that one of the paper’s core contributions is the memorization-aware replay method, but the experimental evidence does not support the claim that “storing high memorization samples is beneficial.” The top-k strategy performs significantly worse than bottom-k and mid-k in nearly all benchmarks, suggesting that storing high-memorization samples is detrimental when the buffer size is small.

The authors agree that high memorization samples are not beneficial in the small buffer regime. They will then focus on claiming that the relative importance grows as memory approaches the “large buffer” regime, and will (i) emphasize the negative results for small buffers, (ii) rephrase the conclusion to highlight a qualitative trend rather than a strong statement, and (iii) add a short discussion on the limited effect sizes in Table 3.

These results also reflect a somewhat lower contribution of the paper, since small buffer size is essential and commonly used in continual learning. The results serve primarily an analytical role, with relatively few methodological improvements, and the contribution is only moderate.

**Reviewer Concerns:**

Almost all reviewer concerns were addressed by the rebuttal. The authors have prepared a comprehensive rebuttal.

**Reviewer Scores:**

I think most reviewers will maintain the scores, owing to the moderate contribution.

---

### Decision · Program_Chairs · 2026-01-26

Reject